# Repurposing Disulfiram as an Antimicrobial Agent in Topical Infections

**DOI:** 10.3390/antibiotics11121752

**Published:** 2022-12-04

**Authors:** Maria Lajarin-Reinares, Eloy Pena-Rodríguez, Mariona Cañellas-Santos, Elisabet Rosell-Vives, Pilar Cortés, Montserrat Llagostera Casas, Maria Àngels Calvo, Francisco Fernandez-Campos

**Affiliations:** 1Topical & Oral Development, Research and Development Reig Jofre Laboratories, 08970 Barcelona, Spain; 2Departament de Genètica i de Microbiologia, Universitat Autònoma de Barcelona, 08193 Barcelona, Spain; 3Biotechnology Department, Research and Development Reig Jofre Laboratories, 08970 Barcelona, Spain; 4Department of Animal Health and Anatomy, Faculty of Veterinary, Universitat Autònoma de Barcelona, 08193 Barcelona, Spain

**Keywords:** disulfiram, skin, *Staphylococcus aureus*, repurposing, DNA arrays, antibiotics, antimicrobial resistance, *Streptococcus pyogenes*

## Abstract

Antimicrobial drugs applied topically offer several advantages. However, the widespread use of antibiotics has led to increasing antimicrobial resistance. One interesting approach in the drug discovery process is drug repurposing. Disulfiram, which was originally approved as an anti-alcoholism drug, offers an attractive alternative to treat topical multidrug resistance bacteria in skin human infections. This study aimed to evaluate the biopharmaceutical characteristics of the drug and the effects arising from its topical application in detail. Microdilution susceptibility testing showed antibacterial activity against Gram-positive bacteria *Staphylococcus aureus* and *Streptococcus pyogenes*. Dermal absorption revealed no permeation in pig skin. The quantification of the drug retained in pig skin demonstrated concentrations in the stratum corneum and epidermis, enough to treat skin infections. Moreover, in vitro cytotoxicity and micro-array analyses were performed to better understand the mechanism of action and revealed the importance of the drug as a metal ion chelator. Together, our findings suggest that disulfiram has the potential to be repurposed as an effective antibiotic to treat superficial human skin infections.

## 1. Introduction

Human skin is one of the first lines of defense against microbial invasion. Healthy skin harbors a diverse range of bacteria, collectively known as the skin microbiome, and depending on the host, bacterial, and environmental factors, this bacterial population may be protective or harmful [1]. *Staphylococcus aureus* and group A streptococci are the two most frequently encountered pathogens causing primary and secondary infections of the skin and skin wounds. Gram-negative organisms such as *Pseudomonas aeruginosa* are sometimes involved in cutaneous infections [2].

Antimicrobial drugs applied topically offer several advantages over systemic administration, including the avoidance of systemic toxicity and side effects, decreased induction of bacterial resistance, high local concentration of antibacterial agents in the infected site, low costs and they also allow the administration of molecules that cannot be given systematically [1,3,4]. However, the widespread use of topical antibiotics (particularly mupirocin and fusidic acid) has led to increasing antimicrobial resistance (AMR) in some settings, making their use extremely difficult and limiting the potential efficacy of such agents [3,4]. Thus, leading to the selection of more resistant *Staphylococcus aureus* strains. Studies conducted in order to better understand the impact of topical antimicrobial use on the development of resistant *S aureus* strains have demonstrated that exposing both MSSA (Methicillin-sensitive *S aureus*) and MRSA (Methicillin-resistant *S aureus*) to fusidic acid and mupirocin resulted not only in the rapid selection of MRSA over MSSA but also the emergence of MDR (Multi-drug resistant) isolates from the original culture [5].

One of the most important human streptococci pathogens of skin is *Streptococcus pyogenes*, which can produce superficial impetigo or more deep-seated cellulitis but also more severe invasive infections such as sepsis, necrotizing fasciitis, and streptococcal toxic shock syndrome [6]. In streptococcal infections, penicillin remains the antibiotic of choice. Importantly, clindamycin-resistant isolates have also been reported in Europe, including France and Finland, and Asian countries, with alarming rates of 94% in China [7]. Other species, such as *Pseudomonas aeruginosa*, are less common but they are a major problem leading to high mortality rates due to the appearance of drug-resistant strains, especially in critically ill and immunocompromised patients [8]. The spectrum of *P. aeruginosa* cutaneous diseases includes localized and often self-limited infections, such as hot-tub folliculitis, chloronychia, hot-foot syndrome and interdigital intertrigo. Ecthyma gangrenosum and sometimes subcutaneous nodules are manifestations in immunocompromised hosts and as such present a medical emergency [7].

Given global concerns regarding antibiotic resistance and relatively limited therapeutic options, especially for some species such as *S. aureus*, the appropriate use of topical agents and the prevention of further resistance are critical [1]. The development of a new antimicrobial is a very slow process and is frequently beset with numerous pitfalls. Repurposing approved drugs is a promising alternative strategy that reduces the time and cost of antibiotic development, as it takes advantage of existing toxicology and pharmacokinetic data from preclinical and clinical trials [9]. 

Disulfiram (D), first commercialized as Antabuse^®^, is a dithiocarbamate that was approved by the US Food and Drug Administration (FDA) in 1951 as a drug to treat alcoholism. It has been widely used in clinics for over 70 years without severe side effects [10]. D is an inhibitor of aldehyde dehydrogenase (ALDH) and it inhibits all the currently identified cytosolic and mitochondrial ALDH isoforms. This fact results in the specific accumulation of acetaldehyde, causing discomfort in patients who drink alcohol as a strategy for their dishabituation [10,11]. 

A cutaneous emulsion consisting of 2% D has been registered in Sweden since 1982 and in Iceland since 1997 for the treatment of scabies and pediculosis in adults and children [12]. Different in vitro and in silico studies have shown that D may be an active substance against the hepatitis C virus, fungi, parasites, bacteria, and cancer [9,10,13,14,15]. A different mechanism of action was proposed to explain these antimicrobial and antiproliferative activities. D has been reported to induce apoptosis, show metal ion-dependent antineoplastic activity and arrest angiogenesis. It has also been shown to inhibit the activating transcription factor/cyclic-AMP-responsive element-binding protein. Moreover, recent studies have shown that D inhibits the activity of the ATPS-binding cassette (ABC) drug transport complex, which is responsible for the development of multiple drug resistance in cancer and fungal infection treatments [16]. D is an electrophile (Figure 1) that readily forms disulfides with thiol-bearing substances. Bacteria possess a diverse range of intracellular cofactors (e.g., coenzyme A), metabolites (e.g., glutathione, mycothiol, and bacillithiol), and enzymes (e.g., thioredoxin) containing thiophilic residues, which D can potentially modify via thiol–disulfide exchange to evoke antimicrobial effects [17]. The use of D as an antimicrobial agent with a different mechanism of action, in comparison with other classical antibiotics, offers an attractive alternative to treating topical, multiresistant bacteria. 

In the context of limited-availability antibiotics and the increase in AMR to them, we focused on repurposing D as an alternative to conventional antimicrobial compounds for superficial human skin infections. Thus, in this study, we present a detailed evaluation of the biopharmaceutical characteristics of the disulfiram drug and the effects derived from topical application.

## 2. Results and Discussion

### 2.1. Antibiotic Susceptibility Testing

*S. aureus*, *P. aeruginosa*, and *S. pyogenes* were selected for susceptibility testing as representative strains related to skin infection and frequently involved with AMR in humans. The minimum inhibitory concentrations (MICs) against these species are summarized in Table 1. As can be seen, D was effective against the Gram-positive bacteria tested, but it did not effectively inhibit *Pseudomonas* spp. growth, a Gram-negative bacterium. *S. aureus* exhibited the greatest susceptibility, with a MIC of 8 µg/mL. This result agreed with those of published reports, where it was found to exhibit a potent activity with the same MIC value [9,17]. In the case of *S. pyogenes,* D had a MIC of 32 µg/mL. This value differs from the previously published value, 16 µg/mL, which could be due to the different strain used, MGAS1882 [18]. Moreover, the MIC result in *P. aeruginosa* correlates well with published data, with a MIC value of more than 64 µg/mL [17]. Conventional antibiotics (clindamycin and gentamicin) were used as controls for the bacteria susceptibility method. Clinical Laboratory Standards Institute (CLSI) guidelines suggest that comparing the MICs of different antibiotics should not be based solely on the numerical value but rather on how far the MIC is from the breakpoint [19]. The CLSI groups MIC values into three categories: susceptible, intermediate, and resistant, based on clinical data and research. Regarding D, there are no defined clinical breakpoints because there are no published data on natural mutants resistant to it, which could be due to the low solubility of disulfiram in aqueous media, preventing experiments from being performed at higher concentrations. Therefore, it was difficult to use the classical approach to compare D activity with other antibiotics. 

The differential effect of D against Gram-positive and Gram-negative bacteria tested could be due to two possible mechanisms of action: Cu^2+^ chelation and compound affinity for the thiol group of cysteine. Proper protein targeting and maturation are fundamental to the homeostasis of all organisms. In Gram-negative bacteria, secreted proteins mature in an oxidative periplasm. Gram-positive bacteria lack this protected compartment due to the absence of an outer membrane. Thus, their secreted proteins must mature in an unregulated environment. Furthermore, cysteine residues perform a variety of essential functions in protein, such as metal binding and enzyme catalysis. However, the reactive thiol groups can also negatively impact the function of a protein by forming incorrect disulfide bonds [20]. The greater exposure of Gram-positive bacteria to D could increase the reactivity of cysteine thiols and, consequently, not allow the correct maturation of the protein, leading to the death of the bacteria. Additionally, the non-effectiveness detected in Gram-negative bacteria can be explained since *P. aeruginosa* is endowed with several mechanisms in the periplasmic space to adapt to copper fluctuations [21].

Moreover, *P. aeruginosa* is a pathogen characterized by the secretion of virulent factors such as the zinc metalloprotease pseudolysisin, also known as LasB or pseudomonas elastase. This virulence factor is highly toxic, causing tissue damage and invasion, processing components of the immune system to cause immunomodulation, and acting intracellularly to initiate bacterial biofilm growth. The inhibition of bacterial virulence factors has gained attention recently as an antimicrobial strategy that is non-destructive to the bacteria. It has been proposed as a second-generation class of antibiotics. By attenuating virulence mechanisms without challenging bacterial cell viability directly, these antibiotic agents would potentially place little or no pressure on the bacterial cell for the emergence of resistant strains [22]. Although disulfiram does not directly reduce the bacterial cell viability, it may reduce virulence factors. Additional studies should be carried out to check this possible effect.

### 2.2. Ex Vivo Skin Absorption Experiment 

To evaluate the permeation profile of D, Franz diffusion cells were used. The use of human skin is usually preferred in these kinds of experiments; however, due to ethical and economic reasons, this tissue is frequently unavailable. In this case, porcine skin is a good surrogate due to its structural similarity to human skin in terms of hair growth density (~20 hairs/cm^2^) and the presence of structures such as Langerhans cells and rete ridges; stratum corneum thickness and contents such as glycosphingolipids and ceramides; stratified, multilayered, keratinizing epithelium; thickness of the viable epidermis (~70 µm); and collagen fiber arrangement in the dermis [23]. Therefore, an ex vivo permeated study with pig skin was used to characterize the absorption of D. In this case, D concentrations were not detected in the chromatogram in any replicate (*n* = 11) at any time in the tested concentration (3.65 mg/cm^2^). This lack of permeability of D in pig skin has also been described in human skin [24]. It is an advantageous property because D can cause adverse effects when it is systemically available due to the inhibition of aldehyde dehydrogenase and dopamine beta-hydroxylase [25].

### 2.3. Quantification of Disulfiram in Skin

Disulfiram, as shown by permeation studies in Section 2.2, was not able to permeate the dermis (D was not detected in the receptor medium in any replicate). However, D could penetrate and be retained in the first layers of the skin (stratum corneum and epidermis). The concentration of the active ingredient in these layers is important to determine whether there are effective concentrations to combat superficial infections. Therefore, the amount of disulfiram retained in pig skin (SC and total skin) was determined. Before the experiment, the density of the skin was calculated. The mean densities of the non-dermatomized and dermatomized pig skins were very similar: 1.05 g/mL and 1.04 g/mL, respectively.

#### 2.3.1. Quantification of D in SC Using the Tape-Stripping Technique 

Before HPLC analysis, we checked the selectivity of the method and the adhesive of the tape used did not produce interferences during the test. The mean amount of D obtained on each strip of the five replicates is shown in Figure 2. A concentration of 48 µg in the SC was obtained (corresponding to 38% of the dose applied in the permeation experiment: 110 µg).

Before calculating the concentration of D in SC, the amount of extracted skin was calculated in each strip group (Table 2) by weighing the strips before and after each extraction. The mean skin retained was 2.20 mg, and the final concentration of D was 20 mg/mL. The concentration obtained was superior to the MIC values found for *S. aureus* and *S. pyogenes*. These results are interesting because superficial skin infections take place in the stratum corneum and epidermis, and this concentration is sufficient to ensure effectiveness. 

#### 2.3.2. Disulfiram Concentration Retained in the Dermatomized Skin 

Over the years, the literature has documented different methodologies for quantifying drug amounts within the skin. The techniques include skin extraction measurements, horizontal stripping and sectioning, the removal of hair follicles, quantitative autoradiography, and spectroscopic methods [26]. Given that our target is the SC and epidermis, the drug content in the epidermis layer was obtained. After tape-stripping the skin, as described in Section 2.3.1, the rest of the skin tissue was cut into pieces of 10 mg, and these were homogenized with mobile phase using the MagNa Lyser instrument. After HPLC analysis, the mean amount detected was 4.32 ± 4.83 µg. Considering the previously obtained skin density (1.04 g/mL), the mean D concentration obtained was 450 µg/mL. This concentration is still higher compared with the MIC value of both microorganisms, confirming the suitability of D for treating skin infections, even if the infection applies to deeper skin layers. 

### 2.4. In Vitro Cytotoxicity Evaluation 

Cytotoxicity was tested in human-transformed keratinocyte (HEK001) cells with the cell proliferation measured by an adenosine triphosphate (ATP) luminescent assay, which is directly related to the presence of metabolically active cells [27]. This method was preferred to other standard methods to determine cell viability, such as the MTT (tetrazolium salt) assay. In this case, MTT was not used because the colorimetric assay is based on the reduction of tetrazolium dye mediated by nicotinamide adenine dinucleotide phosphate (NADPH). D is an aldehyde dehydrogenase inhibitor and it affects NADPH levels. The results with MTT could be inconclusive; therefore, this method was discarded. 

The topical application of active ingredients may cause skin irritation and it is necessary to develop rapid assays to assess potentially damaging effects. An evaluation of the potential for an ingredient or product to cause skin irritation is one of the various studies undertaken in the overall safety assessment process. However, due to increasing concern over animal experimentation, in vitro alternatives must now be developed. Cell cytotoxicity assays are among the most common in vitro bioassay methods used to predict the toxicity of substances in various tissues [28,29]. In this case, we have investigated the cytotoxic effect of D on human keratinocytes for its evaluation potential to cause skin irritation. 

After 72 h of treatment, in a testing range from 0.20 to 33.36 µM, which corresponds from 0.06 to 9.89 µg/mL, the D resulted in a cytotoxic compound with an IC_50_ value of 6.97 µM (2.31 µg/mL) (95% confidence interval (CI), 6.76 to 7.18) (Figure 3). 

We found differences in the cell’s viability compared with other authors. R. Thakare et al. [9] reported an IC_50_ of 250 µg/mL for D using Vero cells and MTT assay. Others such as J. Zha et al. [30] who used Raji and Molt4 cell lines and MTT assay reported an IC_50_ of 0.793 ± 0.08 μM and 1.314  ±  0.229 μM, respectively. The sensitivity of the cytotoxicity assay used differs depending on the different mechanisms, which leads to cell death [31]. Further, as commented before, the aldehyde dehydrogenase inhibition caused by D may interfere with the results of the MTT assay, which could explain these observed differences. E. Pena et al. [32], who tested the same compound (dexamethasone) in two different cell lines HEK001 and HaCaT, reported lower cytotoxicity concentrations for the first cell line but not for the second one. They related these results to the immortalization procedure to obtain the HaCaT, which can modify their cell sensibility to xenobiotics. HEK001 cells are metabolically less active than HaCaT cells and this could mean that they have fewer defense mechanisms against D. Therefore, these considerations could explain the differences detected with R. Thakare. Moreover, Raji and Molt4 are tumor lines and so they have the signaling system altered and their sensibility to xenobiotics is different from other cells’ sensibility. For this reason, it is important to consider what effect is expected in relation to the predicted cell death mechanism.

To better understand the cytotoxicity effect in HEK001 cells, a microarray analysis was performed. 

Even though D was cytotoxic in low concentrations in HEK001 cells, it is important to consider that cytotoxicity tests are useful in screening chemicals for their intrinsic and relative toxicities, but it is impossible to tell whether predictions based on cytotoxicity data alone are sufficiently accurate for labeling and classifying a new chemical according to its likely acute toxicity in vivo [33]. There are difficulties in carrying out a direct extrapolation from the in vitro cell studies to the in vivo topical administration, for example, because cell culture lacks some of the properties of intact skin, such as the presence of stratum corneum and lipids, which limits the permeation of xenobiotics. In a cytotoxicity assay, cells are exposed directly to the active substances. The solubility of D is 4.09 mg/L and it has a log P of 3.88 [34], which gives it poor water solubility; it tends to remain retained between the lipids of the skin, so the D exposed to keratinocytes would be considerably lower than the in vitro concentrations. Further, D has been used for years in humans after oral and topical administration and it is considered a safe product [24]. Thus, although a cytotoxic assay was used as a prescreening tool for skin irritation, due to D characteristics, the cytotoxicity results may be overestimated in comparison with the in vivo possible effect. Further studies are needed to evaluate the consistency and magnitude of possible irritant effects on the skin.

### 2.5. Functional Analysis of Expressed Genes

An expression profile of HEK001 cells after a 48 h treatment with 6.59 µM of D was performed. This concentration was a little bit lower than the D’s IC_50_ and the contact time was a little bit lower than the one used in the cytotoxicity studies so as not to induce significant cell death, which could hide the effect on gene expression. With this concentration, an effect on the cell gene expression was ensured and total cellular mortality was prevented. 

Forty-nine genes presented with a differential expression after treatment with D versus cells without treatment. The differentially expressed genes were further classified into two groups: upregulated genes (Table 3) and downregulated genes (Table 4). The main association between the altered genes is represented in Figure 4. 

As seen in Table 3, there was an upregulated expression of several genes in the samples (28 genes) from the treated cells (HEK001 6.59 µM D after 48 h) in comparison with the non-treated cells (control). Considering that the expression came from cells that have survived the drug exposure, multiple genes related to cell division were overexpressed; they are involved in cell cycle arrest after the detection of DNA damage to ensure its correct repair and they activate signaling pathways as a mechanism of resistance to D. These genes were SGOL2, KDM4D, DNAJC21, MNS1, DOCK11, and DDX5. Additionally, again related to cell division, there was an overexpression of genes such as KIN, which is involved in DNA replication and the cellular response to DNA damage. 

The gene whose expression was most affected was the microRNA (miRNA) CLU|MIR6843. miRNAs were recently discovered to be regulatory expression molecules. Consisting of ≈22 non-coding nucleotides that regulate gene expression via hybridization with messenger RNAs (mRNAs), mRNA degradation and the translational inhibition of targeted transcripts are consequences of this process [35]. In this case, it affected the clusterin (CLU) gene, which has been independently identified as being involved with various fields without clear relationships between them. It is associated with many biological functions, including tissue differentiation and remodeling, membrane recycling, lipid transportation, cell–cell or cell–substratum interaction, cell motility, cell proliferation, and cell death [36]. The affectation in this multifunctional gene could play an important role in the cytotoxicity of D. 

It is important to mention the alteration of the MT1F gene. Metallothioneins (MTs) have a high cysteine residue content, which binds various heavy metals. They are one of the most important biological mechanisms used to protect against toxic metal exposure. Increasing levels of cellular zinc ions activate transcription factor 1 (MTF-1), the major transcriptional regulator [37]. D is a chelator of ions, especially divalent ions such as zinc and copper. D acts as a Cu/Zn ionophore, facilitating cellular accumulation [38]. Therefore, it can alter the cellular levels of these ions and, consequently, produce cell death. Possible mechanisms of MT1F activation of D include induction of zinc influx, generation of reactive oxygen species that induce the release of zinc from MTs, or direct binding to MTF-1 or another protein necessary for MTF-1 transcriptional activation [37]. Moreover, two additional genes related to zinc were altered: ZNF483 and ZNF780A. Both are zinc finger proteins, implicated in transcriptional regulation.

Another important upregulated gene is RASSF6. It is involved in the induction of apoptosis; it may act as a Ras effector protein or suppress the serum-induced basal levels of NF-kB. This result is consistent with the results of other authors, who reported the inhibition of NF-kB and the apoptosis by disulfiram and other carbamates [39,40].

Out of the 49 genes affected, 21 were downregulated (Table 4). An alteration of the expression of genes related to the expression of transmembrane protein, adhesion, and cell migration, such as PLXNB3, KIRREL2, and PLXNB3, was observed. These alterations could decrease the cell’s protection mechanisms and make it more vulnerable to death. Moreover, signaling genes such as RPS6KA1, YWHAG, and ADCY2 also decreased. The chelating effect of D could explain the interruption of signal pathways where divalent ions participate. These results are also in agreement with other authors, who reported the cytotoxicity effect induced by the sequestration and alteration of intracellular and extracellular levels of zinc [41]. 

D has been shown to affect cell ion levels, producing alterations in signaling pathways and cell division. Other authors reported additional pathway alterations such as the inhibition of proteasome signaling [42,43]. Therefore, cytotoxicity was not caused by a single mechanism but by different mechanisms that induce the final apoptosis of the cell. Concerning the mechanism of resistance, metallothionines may have an important role. 

These results suggest that the chelating character of D could be related to its antibacterial effect. For example, Nabil M. Abraham et al. reported that a metal ion chelator can inhibit *S. aureus* biofilm formation and viability and that metallic cations, such as Ca^2+^ and Mg^2+^, play a role in bacterial growth [44]. Nevertheless, additional studies are needed to determine the mechanism of action of D as an antimicrobial. 

## 3. Materials and Methods

### 3.1. Materials

Disulfiram, cocoa butter, stearic acid, and cetostearyl alcohol were kindly gifted by Bioglan AB (Malmö, Sweden). Methanol and KH_2_PO_4_ (Scharlab, S.L., Sentmenat, Spain) were used as mobile phases. Phosphate-buffered saline (PBS) (Sigma-Aldrich, Saint Louis, MO, USA) and hydroxypropyl-beta-cyclodextrin (HPCD) (Pracofar, S.L., Martorell, Spain) were used as receptor media in skin permeation experiments. For cytotoxic evaluation, dimethyl sulfoxide (DMSO) (Sigma-Aldrich, Burlington, MO, USA), keratinocyte serum-free medium, L-glutamine, and epidermal growth factor (hEGF) (ThermoFisher, Waltham, MC, USA) were used. Mueller–Hinton broth (MHB) and chocolate agar were purchased from Condalab (Madrid, Spain). Gentamicin (Sigma-Aldrich, Saint Louis, MO, USA), clindamycin (Sigma-Aldrich, Saint Louis, MO, USA), and lysed horse blood (Themo Fisher, Landsmeer, The Netherlands) were used for antibacterial studies. 

### 3.2. Antibiotic Susceptibility Testing

The strains used in the study were from American Type Culture Collection and were purchased through LGC Standards, S.L.U (Barcelona, Spain). Disulfiram was studied against *S. aureus* (ATCC^®^ 25923), *S. pyogenes* (ATCC^®^ 12344), and *P. aeruginosa* (ATCC^®^ 27853). 

The MIC of D was determined using a broth microdilution assay in 96-well microplates in accordance with method described inCLSI guidelines [19]. MIC was defined as the lowest drug concentration that inhibited visual growth. The overnight culture was adjusted to a 0.5 McFarland and diluted to 1:100 in media; MHB, in the case of *S. aureus* and *P. aeruginosa*, and MHB with 5% LHB for *S. pyogenes,* were treated with serial dilutions of each antibiotic. Gentamicin was used as a positive antibiotic effect control for *S. aureus* and *P. aeruginosa*, and clindamycin was used for *S. pyogenes.* Stock solutions of D (10 mg/mL) were prepared in ethanol. Gentamicin and clindamycin were prepared in the same concentration in each growth medium. D was tested in amounts ranging from 64 to 0.125 µg/mL due to solubility limitations. The final ethanol concentration of these dilutions was below 0.5% to not alter the bacterial growth. Negative controls were used with the respective media and the same ethanol concentration as the samples with D was used. Positive controls were performed with bacteria with the respective media and without D. Gentamicin and clindamycin were tested in ranges from 512 to 0.125 µg/mL. The plates were sealed with adhesive film and incubated at 37 °C for 18–24 h and then the MIC was determined. For each test compound, MIC determination was carried out independently three times. 

### 3.3. Production of Disulfiram Emulsion

The oil phase was a mixture of cocoa butter (7% *w/w*), stearic acid (6% *w/w*), cetostearyl acid (1% *w/w*), and D (2% *w/w*). Fats were melted at 75 °C and the water phase (water q.s 100% with preservative) was heated at the same temperature in a water bath. Then, both phases were mixed and homogenized at 11.000 rpm with an Ultra-Turrax (IKA T-25, Staufen im Breisgau, Germany) for 10 min.

### 3.4. Ex Vivo Skin Absorption Experiment

Abdominal pig skin (*n* = 11) was obtained from a local slaughterhouse (Barcelona, Spain). Permeation studies were accomplished in vertical Franz diffusion cells (VidraFoc, Barcelona, Spain) with a permeation area of 1.54 cm^2^. The skin was cut with a thickness of approximately 0.5 mm using a dermatome CG1371 (Nouvag AG, Goldach, Switzerland). Then, 15% HPCD on PBS at pH 5.5 was used as a receptor medium to maintain sink conditions throughout the experiment, and it was kept at 32 °C and stirred at 500 rpm. A transepidermal water loss (TEWL) measurement device (Delfin technologies, Kuopio, Finland) was used to evaluate skin integrity. Samples from a receptor compartment (300 µL) were taken at fixed times and replaced by an equivalent volume of fresh receptor medium at the same temperature, and they were analyzed using the HPLC method (Waters 2695, Milford, MA, USA) [24]. Sampling times were 1, 2, 3, 4, 5, 20, 21, 22, 23, and 24 h.

The disulfiram emulsion was dosed in a compartment in the equivalent human prescription dose (3.65 mg/cm^2^) [24] to study skin absorption. Placebo formulation was tested as a negative control of permeation and to check the lack of analytical interferences due to the presence of excipients. 

### 3.5. Quantification of Disulfiram in Skin after Permeation Experiments

#### 3.5.1. Determination of Pig Skin Density

Three replicates of each skin were used. The density (ρ) of a solid is the ratio between the mass and the volume it occupies. The volume occupied by the skin was determined using Equation (1):(1)$$\mathrm{Volume}=\frac{M_{1}+M_{2}-M_{3}}{\rho \;}$$
where M_1_ is the weight of the skin, M_2_ is the weight of the flask filled with water, M_3_ is the weight of the flask filled with water and the skin, and ρ is the density of the water (1 g/cm^3^). 

#### 3.5.2. Tape-Stripping Study

After the permeation experiment, five of the eleven Franz cells with the pig skin were used to study the disulfiram retained in the stratum corneum. Prior to tape-stripping, the emulsion residue was removed with a swab soaked in PBS. 

After 24 h, the strips (Tesa^®^ 4101 PV2, Shanghai, China) were carefully adhered to the diffusional skin area and a constant weight (345 g) was used to press the tape onto the skin surface for 10 s. The following tape strip groups were carried out: strip 1, strip 2, strips 3–7, strips 8–12, strips 13–17, and strips 18–20. The tape strips were placed in 50 mL Falcon tubes and we added 4 mL of mobile phase. Then, the samples were sonicated for 15 min in an ultrasonic bath (JP Selecta™, Abrera, Spain) and analyzed using the same HPLC method in Section 3.4. 

#### 3.5.3. Determination of the Concentration Retained in the Dermatomized Skin

After tape-stripping, the skin sections were used to extract the retained D. In total, 10 mg of the area of each replicate per duplicate was cut and introduced into the MagNa Lyser instrument (Roche, Sant Cugat del Valles, Spain) with 600 µL of mobile phase. The homogenization of the tissue was carried out with 5 cycles of 90 s at 6500 rpm. Finally, the samples were analyzed using the HPLC method. 

### 3.6. In Vitro Cytotoxicity Evaluation

HEK001 (CRL2404, ATCC, Manassas, VA, USA) cells were seeded in 96-well plates (*n* = 6) at 10,000 cells/well in 50 µL of medium (keratinocyte serum-free medium, supplemented with 2 mM L-glutamine and 5 ng/mL of hEGF). The cells were incubated at 37 °C in 5% CO_2_ and a 95% air-humidified atmosphere for 24 h. After 24 h of incubation, the cells were treated with 100 µL of D solution in cell media at different concentrations (33.36 to 0.20 µM). A stock solution was prepared in DMSO. The DMSO concentration in the well was below 1% and a DMSO control was carried out. Negative control of cytotoxicity was performed with cells in the same media and positive control of mortality with 1% of SDS (ThermoFisher, Waltham, MA, USA). The exposure period was 72 h.

At the end of the incubation time, a CellTiter-Glo^®^ Luminescent Cell Viability Assay (Dojindo Molecular Technologies, Rockville, MD, USA) was used to evaluate cytotoxicity. CellTiter-Glo^®^ reagent was added (100 µL) directly to every well plate cultured in serum-free medium and incubated for 10 min. Luminescence was measured with a Luminometer Victor X3 (Perkin Elmer, Waltham, MA, USA). Cytotoxicity plots and IC_50_ values were obtained. Cell survival was calculated considering the 100% viability of the untreated control cells (negative control) and 100% of mortality positive control. The percentage of cell viability was calculated using the following equation: (L_Negative control_ − L_Experimental value)_/(L_Negative control_ − L_Positive control_) × 100(2)
where L_Experimental value_ is the luminescence of the sample, L_Negative control_ is the mean luminescence of cells in media control, and L_Positive control_ is the absorbance with 1% SDS control.

### 3.7. Microarrays and Data Analysis

HEK001 cells were seeded in 6-well plates at 250,000 cells/well in 2 mL of medium (keratinocyte serum-free medium and l-glutamine (99:1)). The cells were incubated at 37 °C in 5% CO_2_ and a 95% air-humidified atmosphere for 24 hr. After 24 h of incubation, the cells were treated with 6.59 µM of D. The compound was tested in duplicate and cells with medium without D were used as a control (*n* = 3). The exposure period was 24 h. 

After the incubation time, the total RNA from each sample was extracted and purified from HEK001 cells using a RNeasy^®^ Plus Mini insolation Kit (Qiagen, Hilden, Germany) according to the instructions provided by the manufacturer. Total RNA concentration and quality were spectrophotometrically measured using the absorbance ratio 260:280 nm with NanoDrop^TM^ Lite (ThermoFisher Scientific, Waltham, MA, USA). Once the RNA was obtained, cDNA was synthesized with PxE Thermal Cycler (ThermoFisher, Waltham, MA, USA) according to the manufacturer’s recommendations. For microarray analysis, GeneChip^®^ Clariom S Human Array (ThermoFisher Scientific, Waltham, MA, USA) was used. Data were generated and processed with Affymetrix software (Santa Clara, CA, USA). Gene expression data from the samples were compared using a one-way t-test using stringent transcript cut-off criteria with fold change (FC) > 1.5 and *p*-value ≤ 0.05. 

## 4. Conclusions

Disulfiram is effective against *S. aureus* and *S. pyogenes* and is a potential alternative to classic antibiotics in the treatment of skin infections. Skin permeation studies showed no permeation in pig skin confirming the suitable topical use in superficial human infections. The drug concentration in pig skin tissue was higher than the MIC value of both microorganisms, being potentially effective in the treatment of cutaneous infections. Disulfiram showed cytotoxicity in transformed human keratinocyte cells and several upregulated and downregulated genes after its incubation. The main affected genes were related to cell proliferation signaling and the metabolism of cell ions, which is in agreement with the potential mechanism of action of the drug as a metal ion chelator, essential for cell homeostasis. However, the implementation of D in clinical use has encountered some challenges and further research should be carried out, for example, to evaluate the local tolerance in vivo after D topical administration. Further exploration of the molecular mechanism of disulfiram as an antibacterial agent is needed. Taken together with these results, disulfiram could be a good candidate as an alternative treatment for skin infections, considering the global concern of limited therapeutic options. 

## Figures and Tables

**Figure 1 antibiotics-11-01752-f001:**
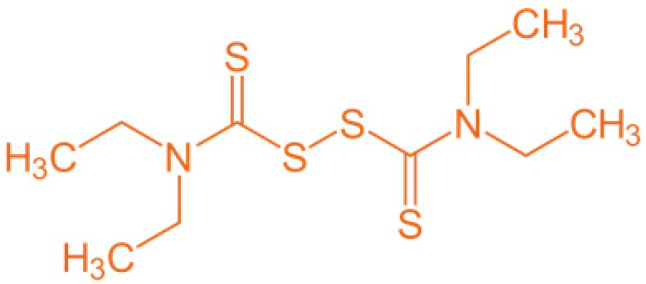
Chemical structure of disulfiram molecule (D).

**Figure 2 antibiotics-11-01752-f002:**
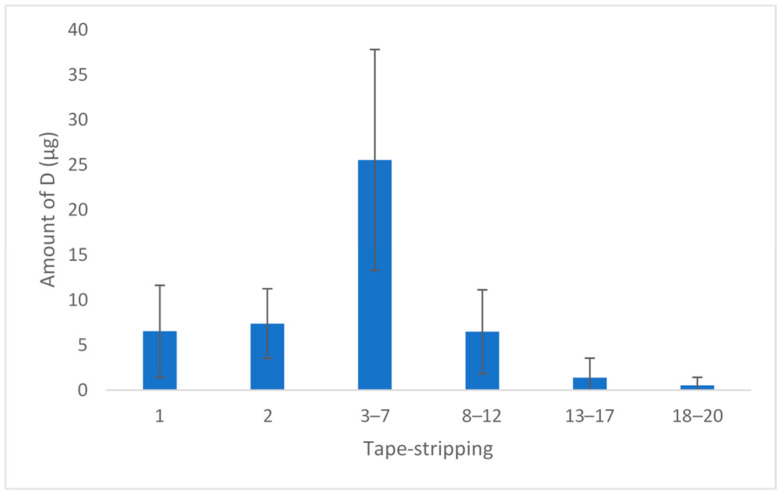
Amount of D per strip (*n* = 5). The results show the mean and standard deviation.

**Figure 3 antibiotics-11-01752-f003:**
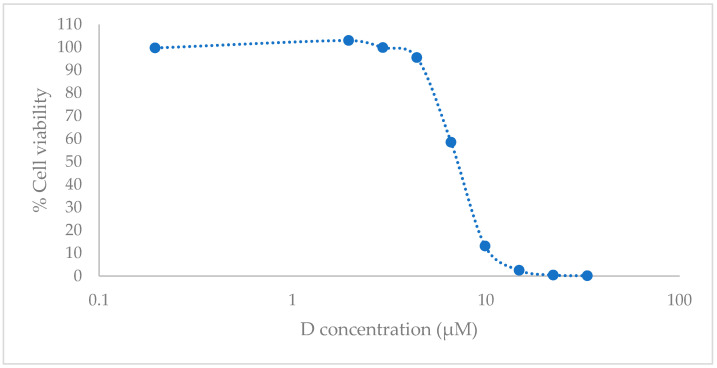
In vitro cell culture studies. HEK001 cell viability with luminometric assay after 72 h post treatment with D. Data are represented as the mean (*n* = 3) of the cell viability percentage, referring to the untreated cell (negative control) and positive control with 1% of sodium dodecyl sulfate (SDS).

**Figure 4 antibiotics-11-01752-f004:**
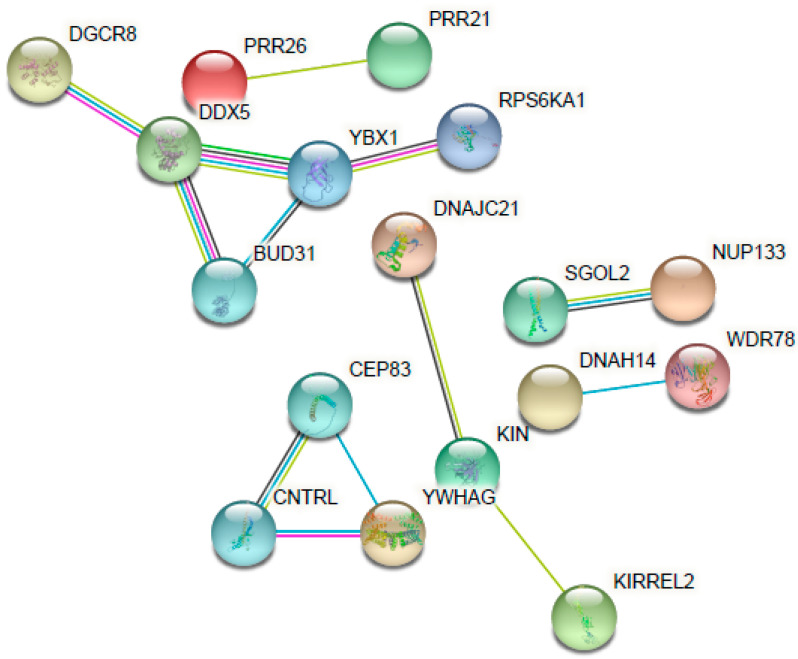
Interaction network of the main altered genes (upregulated and downregulated) for their response to disulfiram. Each gene is represented as a node and each line represents the number of interactions between genes. Image was created using the STRING Database.

**Table 1 antibiotics-11-01752-t001:** Antibacterial activity of disulfiram compared with gentamicin and clindamycin.

Species	MIC					
	Disulfiram		Clindamycin		Gentamicin	
	(µg/mL)	(µM)	(µg/mL)	(µM)	(µg/mL)	(µM)
*Streptococcus pyogenes*	32	108	0.125	1	--	--
*Staphylococcus aureus*	8	27	--	--	0.125	0.262
*Pseudomonas aeruginosa*	>64	>216	--	--	0.250	0.523

**Table 2 antibiotics-11-01752-t002:** Amount of skin retired (mg) for each group strip (*n* = 3) with its standard deviation.

Strip	Mean (mg)	SD
1	0.43	0.26
2	0.39	0.20
3–7	0.68	0.18
8–12	0.39	0.16
13–17	0.23	0.06
18–20	0.07	0.00

**Table 3 antibiotics-11-01752-t003:** Upregulated genes. Target prediction was performed using Affymetrix software. A list of differentially expressed (more than 1.5-fold, *t*-test *p*-value < 0.05) putative targets was obtained. The ratio column corresponds to the fold change in expression for each gene relative to the control, calculated from the normalized values.

Gen Symbol	Gen Description	(D) Raw	(Control) Raw	Ratio
CLU|MIR6843	clusterin|microRNA6843	312.08	106.16	3.08
CCDC138	Coiled-coil domain-containing protein 138	82.23	48.90	1.75
MT1F	Metallothionein-1F	3379.27	2011.00	1.75
DNAH14	Dynein axonemal heavy chain 14	60.23	35.91	1.69
ZNF483	Zinc finger protein 483	77.93	46.19	1.69
WDR78	Dynein axonemal intermediate chain 4	80.69	53.16	1.71
BVES-AS1	BVESantisenseRNA1	59.32	35.94	1.66
ALG6	Dolichyl pyrophosphate Man9GlcNAc2 alpha-1,3-glucosyltransferase	85.31	49.07	1.65
RASSF6	Ras association domain-containing protein 6	52.74	33.33	1.64
DOCK11	Dedicator of cytokinesis protein 11	161.39	107.55	1.64
MNS1	Meiosis-specific nuclear structural protein 1	129.81	88.17	1.61
PAIP2|CTB-43P18.1	Poly(A)bindingproteininteractingprotein2	128.98	84.64	1.57
CNTRL	Centriolin	78.00	48.35	1.59
SGOL2	Shugoshin 2	186.99	117.74	1.59
NUP133	Nuclear pore complex protein Nup133	161.03	99.01	1.57
ZNF780A	Zinc finger protein 780A	105.86	67.45	1.57
CCDC66	Coiled-coil domain-containing protein 66	91.98	57.60	1.59
GLMN	Glomulin	36.51	22.98	1.58
KDM4D	Lysine-specific demethylase 4D	49.39	32.68	1.53
SNX25	Sorting nexin-25	127.76	89.91	1.57
ARHGEF26	Rho guanine nucleotide exchange factor 26	47.67	35.02	1.57
CMC1	COX assembly mitochondrial protein homolog	97.74	62.22	1.53
FASTKD1	FAST kinase domain-containing protein 1	259.25	172.60	1.52
DNAJC21	DnaJ homolog subfamily C member 21	497.80	327.50	1.52
DDX5	ATP-dependent RNA helicase DDX5	142.99	101.83	1.56
CEP83	Coiled-coil domain-containing protein 41	86.58	56.97	1.52
BUD31	Protein BUD31 homolog	29.50	19.38	1.51
KIN	DNA/RNA-binding protein KIN17	131.16	84.77	1.50

**Table 4 antibiotics-11-01752-t004:** Downregulated genes. Target prediction was performed using Affymetrix software. A list of differentially expressed (more than 1.5-fold, *t*-test *p*-value < 0.05) putative targets was obtained. The ratio column corresponds to the fold change in expression for each gene relative to the control, calculated from the normalized values.

Gen Symbol	Gen Description	(D) Raw	(Control) Raw	Ratio
ANKRD36	Ankyrin repeat domain-containing protein 36A	38.41	67.57	2.28
YWHAG	14-3-3 protein gam	25.03	46.32	1.87
EXT1|hunera	Jeck2013ALT	88.64	147.16	1.98
QRICH2	Glutamine-rich protein 2	33.80	65.17	1.82
YBX1	Nuclease-sensitive element-binding protein 1	47.70	69.96	1.68
DGCR8	Microprocessor complex subunit DGCR8	74.69	118.81	1.60
C1orf198	Chromosome1openreadingframe198	43.49	66.29	1.64
FHAD1	Forkhead-associated domain-containing protein 1	46.94	71.21	1.60
PLXNB3	Plexin-B3	56.35	83.87	1.56
CYGB	Cytoglobin	62.20	98.36	1.58
CCDC84	Coiled-coil domain-containing protein 84	167.37	242.60	1.60
F10	Coagulation factor X	70.55	101.19	1.56
PRR21	Proline rich 21	19.56	28.10	1.57
GALNT14	Polypeptide N-acetylgalactosaminyltransferase 14	29.99	44.42	1.55
ADCY2	Adenylate cyclase type 2	31.16	46.58	1.55
KIRREL2	Kin of IRRE-like protein 2	27.16	39.19	1.54
RPS6KA1	Ribosomal protein S6 kinase alpha-1	34.27	48.01	1.52
AP1M1	AP-1 complex subunit mu-1	142.99	204.18	1.53
PRR26	Proline-rich protein 26	40.47	60.82	1.50
CRYM	Thiomorpholine-carboxylate dehydrogenase	34.46	52.06	1.51
WNT8B	Wingless-type mmtv integration site family	20.00	26.53	1.50

## Data Availability

Data are available upon request due to intellectual property.

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
