# Peer review of "Repurposing Disulfiram as an Antimicrobial Agent in Topical Infections"

_antibiotics, 2022, doi:10.3390/antibiotics11121752_

Round 1

Reviewer 1 Report

 This manuscript by Maria Lajarin-Reinares et al describes the use of the anti-alcoholism drug  ‘Disulfiram’ in treating bacterial infections in farm animals. The study has evaluated the minimum inhibitory concentrations (MICs) of Disulfiram, Clindamycin, and Gentamicin against gram-positive and gram-negative bacteria. The skin absorption experiment demonstrated that the drug is impermeable to pig skin and for bovine udder skin, the drug is permeable in few replicates.

By histological examination, the authors showed that the reduced thickness of the stratum corneum (SC) layer and more hair follicles of the bovine udder skin are responsible for the permeability of the drug and therefore raising a question on its use in the cows. The authors further showed the cytotoxicity of the drug in HEK001 cell lines and performed microarrays to identify up-and-down-regulated genes after drug treatment.

Author Response

Dear reviewer, 

Thanks for your comments. We greatly appreciate them and we are pleased that you have positively valued our manuscript. 

Best regards, 

Maria 

Reviewer 2 Report

Maria et al. showed chelating character of D could be a major factor in antibacterial effects against different gram positive and negative bacteria. I have few queries regarding this study,

1.     As Disulfiram (D) is less affective against p. aeruginosa, I suggest checking its effectiveness against different virulent factors like Pyoverdine and pyocyanin released by the bacteria involved in bacterial virulence and pathogenesis as well. Moreover, LasA/B Protease-elastase also has an ability to produce corneal ulcers, necrotic skin lesions and pulmonary hemorrhages. LasB elastase is a zinc metalloprotease capable of destroying or inactivating a wide range of biological tissues and immunological agents. Therefore, rather than direct killing efficiency of D against P. aeruginosa it might reduce the release of virulent factors. It also can provide the question against insolubility issue. These virulent factors given as an example; other factors might be targeted. Before going into its detailed study like drug permeability in different skin and contact with circulation I suggest showing more data (mechanism of action) about the potential of D against bacteria. 

2.     Cytotoxic study is not clear enough to address its applicability. My also concern about the toxic effect of D on peripheral blood cells as it might penetrate (although authors showed no penetrated drug in circulation) and comes in a contact with circulation. If it happens then I suggest checking the effect of D on human peripheral cells like PBMC.

Author Response

Dear Sir/Madame,

Firstly, thank you for your comments and the opportunity to revise our manuscript.’ The suggestions offered, have been immensely helpful. We have included your comments and responded to them individually in red color. The changes are made by control change and the revised manuscript is attached. Please, noted that the focus of our articles has changed, Disulfiram is proposed for human topical infections

We hope the revised manuscript will better suit the and we thank you for your continued interest in our research.

  1. As Disulfiram (D) is less affective against p. aeruginosa, I suggest checking its effectiveness against different virulent factors like Pyoverdine and pyocyanin released by the bacteria involved in bacterial virulence and pathogenesis as well. Moreover, LasA/B Protease-elastase also has an ability to produce corneal ulcers, necrotic skin lesions and pulmonary hemorrhages. LasB elastase is a zinc metalloprotease capable of destroying or inactivating a wide range of biological tissues and immunological agents. Therefore, rather than direct killing efficiency of D against P. aeruginosa it might reduce the release of virulent factors. It also can provide the question against insolubility issue. These virulent factors given as an example; other factors might be targeted. Before going into its detailed study like drug permeability in different skin and contact with circulation I suggest showing more data (mechanism of action) about the potential of D against bacteria. 

Thank you for pointing this out, that’s a fantastic idea and it could be helpful to a better comprehension of the drug and its mechanism effect. Despite it would have been interesting to explore this aspect, this doesn’t be the objective of our study. We had two main objectives, on the one hand, we wanted to focus on the total lethality of Disulfiram following the method described in CLSI and on the other hand, studying the suitability of a topical application of this active ingredient. Moreover, there are other articles regarding the antimicrobial characterization of the active already published, but its possible effects derived from the clinical application, not. For this reason, we have highlighted more the second part. In fact, as we mentioned in page 3 line 138 the aim of the study is present a detailed evaluation of the biopharmaceutical characteristics of the disulfiram drug and the effects derived from a topical application. However, your interesting suggestion is incorporated in the section 2.1 (page 4 line 183). Additional future studies regarding virulence factor will be consider as part of my thesis. 

  1. Cytotoxic study is not clear enough to address its applicability. My also concern about the toxic effect of D on peripheral blood cells as it might penetrate (although authors showed no penetrated drug in circulation) and comes in a contact with circulation. If it happens then I suggest checking the effect of D on human peripheral cells like PBMC.

We appreciate the reviewer’s feedback; however, we consider not study the toxicity of D on peripheral blood cells because D is an active approved by FDA since 1951 (page 3, line 112) and no hemolysis adverse effect have been described. Moreover, the drug does not permeate the systemic circulation (there were found no concentration in the receptor compartment of the Franz diffusion cells) and our objective is to study the cytotoxicity on the skin (place where our compound is accumulated). For this reason, the keratinocytes cells were selected. Potential skin irritation by an ingredient or product is one of the various studies undertaken in the overall assessment process. This approach to detect skin irritation is widely used in cosmetic and pharmaceutical industry employing cell culture as an alternative method to in vivo testing. Cell cytotoxicity assays are among the most common in vitro bioassay methods to predict the toxicity of a wide range of substances. Since it may not be clear from the manuscript, this additional information is included (section 2.3).

Reviewer 3 Report

This is a good-written manuscript on a topic that is relevant. I have no serious or substantive comments.

I have a few suggestions on how to improve the manuscript:

1. “in vitro/in silico” should be in italic

2. Keywords: resistance? maybe better use “AMR” or “antimicrobial resistance.”

3. Table 1. I recommend you represent data in μg/mL and in mmol/μmol at the same time, based on Disulfiram/Clindamycin/Gentamicin Molar mass

4. Table 2. What is SD? Is it standard deviation? If so, you should use another statistical method because SD could not be almost as mean, or in case Dif (1/h) SD bigger than the mean. Maybe you should represent data with 25/75 percentile.

Reviewer 4 Report

Review report is attached.

Round 2

Reviewer 2 Report

I appreciate for addressing my concerns. I recommend the paper for publication.

Author Response

Dear reviewer, 

Thanks for your comment. We greatly appreciate it and we are pleased that you have positively valued our manuscript. 

Best regards, 

Maria 

Reviewer 3 Report

1. One of the keywords, “arrays”, should it be “DNA arrays”? Maybe just “arrays” is ok, but please check it.

2. write “in vivo” in italic in the conclusion

Author Response

Dear reviewer, 

Thanks for your comments. Revised accordingly. The change is included in the manuscript.

Best regards, 

Maria 

Reviewer 4 Report

Thank you addressing significant corrections. Please see the attached report V2

Author Response

Dear reviewer, 

Thanks for your comments. Revised accordingly. The change is included in the manuscript.

Regarding the use of pig skin for the study of D permeation, comment that pig skin is a  widely accepted model to predict results in humans as a first approximation (https://doi.org/10.1016/j.tiv.2008.10.008, https://doi.org/10.3390/pharmaceutics12020112). Also, as we mentioned in line 208, other studies of D as same concentration does not show permeation in human skin. For this reason, we performed the extrapolation. Nevertheless, we wanted to be cautious and as we mentioned in line 636, further research should be done, for example to evaluate the local tolerance in vivo after D topical administration.

Best regards, 

Maria